# NAA10 as a New Prognostic Marker for Cancer Progression

**DOI:** 10.3390/ijms21218010

**Published:** 2020-10-28

**Authors:** Sun Myung Kim, Eunyoung Ha, Jinyoung Kim, Chiheum Cho, So-Jin Shin, Ji Hae Seo

**Affiliations:** 1Department of Gynecology and Obstetrics and Institute for Cancer Research, School of Medicine, Keimyung University, Daegu 42601, Korea; sun7@snu.ac.kr (S.M.K.); c0035@dsmc.or.kr (C.C.); 2Department of Biochemistry, School of Medicine, Keimyung University, Daegu 42601, Korea; eyha@dsmc.or.kr; 3Department of Internal Medicine, School of Medicine, Keimyung University, Daegu 42601, Korea; takgu@dsmc.or.kr

**Keywords:** acetyltransferase, biomarker, cancer prognosis, NAA10

## Abstract

N-α-acetyltransferase 10 (NAA10) is an acetyltransferase that acetylates both N-terminal amino acid and internal lysine residues of proteins. NAA10 is a crucial player to regulate cell proliferation, migration, differentiation, apoptosis, and autophagy. Recently, mounting evidence presented the overexpression of NAA10 in various types of cancer, including liver, bone, lung, breast, colon, and prostate cancers, and demonstrated a correlation of overexpressed NAA10 with vascular invasion and metastasis, thereby affecting overall survival rates of cancer patients and recurrence of diseases. This evidence all points NAA10 toward a promising biomarker for cancer prognosis. Here we summarize the current knowledge regarding the biological functions of NAA10 in cancer progression and provide the potential usage of NAA10 as a prognostic marker for cancer progression.

## 1. Introduction

Cancer is currently the second leading cause of death worldwide following heart disease [1]. With the rapidly advancing biomedical technologies, discovery of biomarkers for early detection and progress of cancer has been the focus of intense research [2].

N-terminal acetyltransferase (NAT) is an acetyltransferase that targets the N-terminal α-amino group of nascent proteins. NAA10, a catalytic subunit of NatA, also functions as lysine acetyltransferase (KAT) that acetylates internal lysine residues of proteins [3]. Accumulating evidence demonstrated that NAA10, both via NAT and KAT activities, plays key roles in regulating tumorigenesis processes such as cellular proliferation, apoptosis, migration, and autophagy. Evidence also demonstrated that NAA10 is highly upregulated in various malignancies, including breast, bone, colorectal, liver, lung, and prostate cancers, and that expression level of NAA10 is correlated with the cancer progression, an implication for possibility of NAA10 as a cancer biomarker [4,5,6,7,8,9]. Based on above referenced studies, this review discusses the possibility of NAA10 as a prognostic cancer biomarker and describes the biological functions of NAA10 in cancer progression.

## 2. NATs

Most proteins undergo one or more types of modifications to form stable structure and/or maintain catalytic activities. More than 200 different types of protein modification, ranging from small chemical modifications—acetylation and phosphorylation—to small molecule bindings—ubiquitination and sumoylation—occur in the cell [10,11]. N-terminal acetylation (Nt-acetylation) catalyzed by NAT, is the attachment of an acetyl group from acetyl-CoA to the α-amino group of N-terminal residues of newly synthesized polypeptides (Figure 1A). It is a representative process of co-translational protein modifications in eukaryotes and affects more than 80% of all human proteins [12]. Nt-acetylation of proteins regulates various cellular events, including protein–protein interaction, subcellular localization, aggregation and folding, and protein turnover [13,14,15,16]. Along with Nt-acetylation, acetylation of the ε-amino group of an internal lysine residue of protein, mediated by KATs, also frequently occurs in the cell (Figure 1B) [17].

To date, NAT family in eukaryotes comprise eight isoforms, from NatA to NatH. NatA, B, C, and E are composed of a unique catalytic subunit and auxiliary subunits and the others have one catalytic subunit. Subcellular localization of NATs varies from cytosolic and ribosome-associated (NatA–NatE) areas to Golgi membrane (NatF), organelle lumen (NatG), and cytosolic but non-ribosomal (NatH) area [3]. The substrate specificities of NAT complexes toward different proteins are determined by the identity of the first two amino acids. NatA, a major NAT that acetylates about 40% of the human proteome, is composed of a catalytic subunit NAA10 and an auxiliary subunit NAA15. NatA co-translationally acetylates the N-terminus of small amino acids (Ala, Cys, Ser, Gly, Thr, and Val) that are exposed after methionine cleavage by a methionine aminopeptidase [3,18].

## 3. NAA10

NAA10 is a catalytic subunit of the NatA complex that acetylates the N-terminus of proteins following aminopeptidase mediated methionine cleavage [19]. NAA10 is a human orthologous gene of arrest-defective 1 (ARD1), which was first identified in *Saccharomyces cerevisiae* by Whiteway and Szostak in 1985 [20]. In humans, NAA10 is located on chromosome Xq28 and composed of eight exons, which is highly conserved across organisms from yeasts to mammals [21].

NAA10 acetylates the ε-amino group of lysine residues of the substrate proteins. The substrates of NAA10 include androgen receptor (AR), Runt-related transcription factor 2 (Runx2), heat shock protein 70 (Hsp70), and phosphoglycerate kinase 1 (PGK1) [22,23,24,25]. DePaolo et al. reported the function of NAA10 in the tumorigenesis of prostate cancer. They demonstrate that NAA10 acetylates AR at Lys618, an event that dissociates AR from AR-HSP90 complex and translocates AR into nucleus where AR activates its target gene expressions and stimulates the androgen-dependent tumorigenesis [22]. NAA10 also regulates bone formation. Yoon et al. reported that NAA10 acetylates Lys225 residue of Runx2, a modification that inhibits Runx2-mediated gene transcription and regulates the differentiation of osteoblast differentiation in the bone [23]. More recently, Seo et al. revealed a role of NAA10 in the stress response. They unraveled that NAA10 acetylates the Lys77 residue of Hsp70 in response to cellular stress and this modification is attributable to the maintenance of protein homeostasis and cell survival under the stress conditions [24]. Based on the evidence reported for decades, it has been established that NAA10 plays important roles to regulate cell proliferation, differentiation, and survival by acetylating its target proteins.

With regard to the molecular structure, NAA10 consists of 235 amino acids and N-terminal region of NAA10 is critical to form the NatA complex with NAA15 [26]. Acetyltransferase domain is located between amino acids 45 and 130, containing an acetyl-CoA binding site (Figure 2). Intriguingly, NAA10 is found both in the cytosol and in the nucleus, whereas NAA15 is localized only in the cytosol, implying a unique function of NAA10 in the nucleus independent of NAA15 [26]. Park et al. showed that NAA10 has a nuclear localization signal (NLS) in ATD and that NLS-deleted NAA10, which cannot enter the nucleus, resulted in cell morphological changes and growth impairment, demonstrating a critical role of NAA10 translocation into nucleus in cell cycle progression [27]. On the basis of its KAT activity, NAA10 could possibly be involved in epigenetic regulation of gene expression. Another NAT family, including NAA20, NAA30, NAA40, and NAA50, are also observed both in the cytosol and in the nucleus, however, their translocations or cellular functions in nucleus have not yet been investigated [28].

In contrast to N-terminal region of NAA10, C-terminal region is predicted as a highly intrinsically disordered region (IDR) which lacks a fixed three-dimensional structure, allowing NAA10 to interact with different proteins with different consequences, implying that structural and functional properties of NAA10 might be decided by diverse protein modifications and formation of protein complexes, which is essential to conduct a proper function in a specific cell type under a certain condition [29].

Indeed, previous studies reported several post-translational modifications of NAA10 that occur under the specific condition and regulate the enzymatic activity and cellular function of NAA10. Autoacetylation of NAA10 at K136 residue, which is essential for enzymatic activity of NAA10, is rapidly stimulated by anticancer drug treatment, leading to the activation of cellular stress response and the protection of cancer cells against cell death [30]. Under glutamine deprivation or hypoxic condition, mTOR-mediated NAA10 phosphorylation at S228 residue is downregulated, resulting in the activation of an acetyltransferase activity of NAA10 and the promotion of protective autophagy processes in cancer cells [25,31]. Phosphorylation also occurs at S209 residue of NAA10 by IKKβ, which induces proteosomal degradation of NAA10 [32]. Under normoxic condition, NAA10 is hydroxylated by factor inhibiting HIF (FIH) at W38 residue, leading to opening the gate at the catalytic pocket of NAA10 that allows the lysine acetylation of substrate protein [33]. In addition to these sites, bioinformatics data (Uniprot: P41227) uncovered that NAA10 has at least additional five post-translational modification sites in C-terminal region of NAA10, including S182, S186, S205, S213, and S216 (Figure 2). Furthermore, NAA10 is also observed to be cleaved after anticancer drug treatment, although the biological meaning of this event needs to be elucidated [26].

Previous studies also support the importance of interacting protein of NAA10 for its functional properties. In vitro, purified NAA10 recombinant is prone to aggregate then easily loses its catalytic activity, suggesting that stability and enzymatic activity of NAA10 protein rely on its interaction with other proteins [34]. Crystal structure analysis of NAA10 revealed that binding of NAA10 to NAA15 induces a conformational change and increases the NAT activity of NatA complex [35]. Enzymatic and physical properties of NatA complex are also regulated by the interaction with an intrinsically disordered Huntingtin yeast two-hybrid protein K (HYPK) [36]. All these results suggest that the physiological roles of NAA10 could be dynamically and transiently regulated by protein modifications and interactions over time and space, which should not be overlooked to understand the physiological roles of NAA10 in cancer.

## 4. Expression of NAA10 in Cancer

N-terminal acetylation occurs co-translationally over 80% of the human proteome. Due to its essential role in protein synthesis, NAA10 is expressed in a wide range of cell types. N-terminal acetylation of newly synthesized proteins might be especially important in rapidly dividing cells. On the basis of this reason, it is reasonable to expect that the role of NAA10 is critical for cancer cells. Accordingly, the expressions of NAA10 are highly upregulated in various types of cancer tissues compared to its adjutant normal tissues.

### 4.1. Breast Cancer (BCa)

NAA10 has been implicated in the oncogenesis of BCa. BCa-derived tissues exhibited elevated expressions of NAA10. Wang et al. analyzed 356 clinical breast specimens and confirmed that the expression levels of NAA10 in cancerous tissues were upregulated compared with those in non-cancerous tissues [4]. They also demonstrated that higher expression level of NAA10 is correlated with the degree of cancer invasiveness and metastasis, an implication that NAA10 may be a potential prognostic biomarker for monitoring the progress of BCa. Of note, contrary to Wang et al., Kuo et al. reported a possible function of NAA10 as a tumor suppressor in BCa. They showed that high expression of NAA10 was associated with better clinical outcomes for patients with BCa [37]. This issue will be discussed further later in this review.

### 4.2. Lung Cancer (LCa)

Overexpression of NAA10 in LCa has been reported. Lee et al. reported upregulated expression of NAA10 in LCa and its correlation with the cancer progression [8]. They also showed the oncogenic effect of NAA10 in vitro. They proved that NAA10 interacts with DNA methyltransferase 1 (DNMT1) thereby silencing E-cadherin, a tumor suppressor gene. Hua et al. reported a contradicting result that showed downregulated expression of NAA10 in LCa, proposing tumor-suppressive effect of NAA10 [38].

### 4.3. Hepatocellular Carcinoma (HCC)

HCC, the most frequent representative malignancy of the liver, is the second leading cause of cancer-related deaths in East Asia and the third in western countries [39]. Similar to many other malignancies, the overexpression of NAA10 has been observed in HCC. Shim et al. investigated the role and clinical involvement of NAA10 in HCC development [7]. They measured intratumoral NAA10 mRNA levels in patient-derived HCC tissues and found that the high transcription level of NAA10 mRNA was closely related to HCC progression. Lee et al. added the same finding by showing proportionally increased levels of NAA10 from low grade dysplasia to HCC and a correlation between NAA10 expression and HCC progression [19].

### 4.4. Colorectal Cancer (CRC)

CRC, a malignancy of the inner lining of the colon or rectum, is the third leading cause of cancer-related mortality worldwide [40]. Similar to the results above, increased expressions of NAA10 have been observed in CRC. Ren et al. showed increased mRNA and protein expressions of NAA10 in CRC tissues [41]. Jiang et al. and Yang et al. reported high expression levels of NAA10 in patients with CRC. These results suggest the potential role of NAA10 as a prognostic biomarker for CRC [6,42].

### 4.5. Osteosarcoma

NAA10 is known to take part in embryogenesis and regulate bone formation. Consequently, its dysregulation causes severe developmental defects and bone cancer [43,44]. A recent report showed an association of the higher transcription level of NAA10 with several types of bone cancer [5].

### 4.6. Oral Squamous Cell Carcinoma (OSCC)

OSCC is a major malignancy in the oral capacity, comprising approximately 90% of all oral neoplasms [45]. Few studies reported the role of NAA10 in OSCC. Zeng et al. studied the involvement of NAA10 in the OSCC and found that NAA10 expression levels were highly increased in 98 out of 124 OSCC specimens [46].

### 4.7. Prostate Cancer (PCa)

PCa, one of the most common malignancies in men, gives rise to the second leading cause of cancer-related deaths worldwide [47]. The progression of PCa is highly dependent on AR signaling, a pathway that is essential for the development and function of the prostate gland [48]. Patients with PCa show significantly elevated expression levels of AR. Additionally, many studies showed that AR is causative in the pathogenesis and the progression of PCa [49]. Of a couple of post-translational modifications that modulate the activities of AR, acetylation activates AR and AR dependent signaling pathway ultimately leading to the development and the progression of PCa [50]. Given the function of NAA10 in the pathogenesis of various cancers, Wang et al. showed that NAA10 expressions were highly increased in PCa [9]. They conducted immunohistochemistry (IHC) on 64 PCa tissues and found substantially higher levels of NAA10 in PCa tissues than in adjacent normal tissues [9]. Furthermore, DePaolo et al. confirmed that NAA10 forms NAA10-Hsp90-AR complex and acetylates Lys618 residue of AR. This acetylation allows AR to translocate into the nucleus and sequentially stimulates the expression of AR target genes required for the progression of PCa [22]. These results suggest the possible usage of NAA10 as a novel biomarker for PCa.

## 5. NAA10 as a Prognostic Marker

Innumerable studies have demonstrated that the expression level of NAA10 in tumor tissues is closely correlated with disease progression and clinical outcomes of various cancers (Table 1).

### 5.1. NAA10 and Cancer Survival

#### 5.1.1. BCa

NAA10 appeared to be a promising biomarker for assessment of prognosis after postoperative chemotherapy. Wang et al. showed that NAA10 was positive up to 50/82 (61.0%) at primary diagnosis in breast invasive ductal carcinomas (IDC) specimens. Of 50 patients with positive NAA10, 29 patients recurred and underwent a second surgery [4]. From these observations, they suggested that the elevated NAA10 protein is associated with poor prognosis in BCa patients, and thus that NAA10 may be utilized for the prediction of prognosis. Caution should be taken since the data are controversial. Kuo et al. found higher levels of NAA10 transcript in BCa tissues from longer relapse-free surviving patients compared with those from shorter relapse-free surviving patients [37]. In addition, Zeng et al. showed that highly expressed NAA10 was associated with better survival rate in BCa patients [51]. These reports suggest that NAA10 expression in BCa correlates positively with cancer survival.

#### 5.1.2. LCa

Lee et al. showed that NAA10 overexpression is associated with poor survival of LCa [8]. They performed IHC of NAA10 expression from 90 patients with lung adenocarcinoma and found that 48 patients with high levels of NAA10 showed poorer survival rates compared to 42 patients with low levels of NAA10. These results indicate that NAA10 was overexpressed in more than half of 90 LCa tissues and that NAA10 may play a role in the progression of LCa. However, as in the case of BCa, the other study has presented an opposite result regarding the oncogenic role of NAA10. Hua et al. reported that the higher expression of NAA10 is correlated with better survival of female patients with adenocarcinoma, suggesting the suppressive role of NAA10 in LCa progression [38].

#### 5.1.3. HCC

HCC is characterized by high mortality and poor survival rate. Lee et al. showed that NAA10 is highly upregulated in HCC tissues and that NAA10 overexpression was associated with microvascular invasion, poor tumor differentiation, and poor survival rate [19].

#### 5.1.4. CRC

Jiang et al. demonstrated that, of 106 patients with high expression of NAA10, 74 (69.8%) died of cancer-related causes, as opposed to 7 (41.1%) out of 17 patients with low NAA10 [6]. They revealed a significantly shortened overall survival in patients with NAA10 positive expressions.

#### 5.1.5. Osteosarcoma

Osteosarcoma patients with more than average levels of NAA10 expression showed significantly shorter overall survival, an implication that enhanced NAA10 expression is associated with poor prognosis [5].

#### 5.1.6. OSCC

A study that analyzed the survival of OSCC showed that patients with positive NAA10 expression had a better overall survival rate than those with negative NAA10 expression, a statement that high level of NAA10 in OSCC is correlated with the better prognosis [46].

### 5.2. Invasion and Metastasis

#### 5.2.1. BCa

As previously mentioned, Wang et al. highlighted the correlation of NAA10 expression with BCa progression [4]. They showed that the level of NAA10 protein in breast carcinoma patients was distinctly related to lymph node metastasis, with 94.0% (47/50) of metastatic tumor showing increased expression, as compared to 6.0% (3/50) of non-metastatic tumors. Conversely, other studies have emphasized that NAA10 expression levels are negatively associated with lymph node metastasis in BCa patients [37,51]. Kuo et al. found that NAA10 transcription levels were higher in patients with fewer lymph node metastases than in those with more lymph node metastases [37]. Zeng et al. also showed that NAA10 levels were higher in patients with fewer lymph node metastases [51].

#### 5.2.2. LCa

There are no reports of whether high levels of NAA10 expression are positively correlated with LCa invasion and metastasis. However, Hua et al. found lower expression of NAA10 in malignancies with lymph node metastasis compared to non-lymph node metastasis [38]. They showed significantly decreased levels of NAA10 in 13 out of 15 lung cancer patients with positive lymph node metastasis compared with those with negative lymph node metastasis, suggesting the suppressive role of NAA10 in the tumor metastasis.

#### 5.2.3. HCC

The high level of intratumoral NAA10 mRNA has been implicated in the process of microvascular invasion in HCC patients. Shim et al. investigated the role and clinical involvement of NAA10 in the development of HCC [7]. They measured intratumoral NAA10 mRNA levels in patient-derived HCC specimens and observed that patients with higher expression level of NAA10 showed more frequent microvascular invasions than patients with lower expression levels of NAA10.

#### 5.2.4. Osteosarcoma

Chien et al. observed higher expression level of NAA10 in osteosarcoma patients with metastasis compared with those without metastasis [5]. They showed that NAA10 interacts with matrix metalloproteinase-2 (MMP-2) via its acetyltransferase domain and in turn promotes the stabilization of MMP-2 protein, leading to the increases in cell invasion and tumor metastasis.

#### 5.2.5. OSCC

NAA10 was proposed as a tumor suppressor in OSCC. Higher expression levels of NAA10 negatively correlated with lymph node metastasis in OSCC [46].

### 5.3. Recurrence

Not sufficient studies as to the correlation between the expression level of NAA10 and cancer recurrence have been conducted. Wang et al. reported recurrence after postoperative chemotherapy in 29 breast cancer patients with high levels of NAA10 [4]. The expression levels of NAA10 in HCC were positively correlated with tumor recurrence [19]. However, inverse correlations were revealed between NAA10 expression and the recurrence of OSCS [46].

## 6. NAA10 in Tumorigenesis

As mentioned above, several research groups have demonstrated that the expression levels of NAA10 in cancer tissues are considerably correlated with the progression, metastasis, and the survival. However, the roles of NAA10 in cancer tissues seem to be very complicated because the overexpression of NAA10 is closely associated with poor outcomes in HCC and CRC, but with better outcomes in BCa and OSCC. Mechanistic studies have revealed diverse molecular mechanisms of NAA10 and defined NAA10 as a double-faced player, an oncoprotein, and a tumor suppressor. As mentioned before, NAA10 has an intrinsically disordered structure, suggesting the existence of numerous kinds of protein complexes that are functionally different and specific for space–time context. NAA10 is reported to have 95 kinds of physical interactions in databases (Uniprot: P41227), however, the physiological meaning and regulation of these interactions are mostly unknown. This review introduces a part of them based on the functional analyses by biochemical and cellular approaches (Table 2).

### 6.1. NAA10 as an Oncoprotein

The oncogenic properties of NAA10 have been firmly established. Overexpression of NAA10 is associated with microvascular invasion, lymph node metastasis, and lower survival rates in various malignancies, including BCa, LCa, HCC, and osteosarcoma [17]. These findings surely imply the role of NAA10 as an oncoprotein.

Molecular mechanistic studies have revealed that NAA10-mediated protein acetylation plays a crucial role in regulating cellular events that are significant for cancer development, such as cell cycle progression, cell death, migration, and autophagy. In LCa cells, NAA10 acetylates and activates β-catenin, an event that enhances the expression of cyclin D1 protein and then leads to the uncontrolled cell proliferation [56]. NAA10 acetylates and stabilizes phosphatase Cdc25A [52]. The stabilized Cdc25A dephosphorylates cyclin-Cdk complexes and consequently promotes cell proliferation. Furthermore, NAA10 suppresses the tumor suppressor gene E-cadherin by regulating DNMT1 in LCa cells [8]. NAA10 recruited DNMT1 to the E-cadherin promoter independent of acetyltransferase activity, suggesting acetylation-independent oncogenic potential of NAA10. Recently, Vo et al. showed that NAA10-mediated acetylation of aurora kinase A (AuA) promotes proliferation and migration of BCa cells, indicating the probable role of NAA10 in cancer development [54]. As for apoptosis, Park et al. described cancer cell survival mechanisms mediated by Hsp70 acetylation under stress conditions [53]. NAA10-mediated Hsp70 acetylation inhibits cell death by preventing apoptotic protease-activating factor-1 (Apaf-1) and apoptosis-inducing factor (AIF)-controlled apoptotic processes. For the autophagy process involved in brain tumor progression, Qian et al. showed that NAA10 acetylates Lys388 residue of PGK1 under glutamine deprivation [25]. Acetylated PGK1 binds to and phosphorylates Beclin1 and induces in glioblastomas. These results suggest that NAA10 could be a promising candidate for cancer biomarker.

### 6.2. NAA10 as a Tumor Suppressor

As previously described, NAA10 can also act as a tumor suppressor in various malignancies, including BCa, LCa, and OSCC [37,38,46]. In those malignancies, higher expression levels of NAA10 are correlated with better clinical outcomes; better survival, smaller tumor volume, and lower rates of lymph node metastasis.

New insights into the molecular mechanisms of how NAA10 reduces tumor proliferation and metastasis have been discovered. Kuo et al. identified a tumor suppressive activity of NAA10 in BCa, in which NAA10 retarded cancer cell growth and stimulated autophagy by inhibiting mTOR signaling [37]. Regarding the cellular motility, NAA10 directly acetylates the Lys608 residue of myosin light chain kinase (MLCK) and inhibits MLCK activity required for cell migration and expansion [55]. Hua et al. have depicted the molecular mechanism by which NAA10 interrupts cancer cell metastasis in LCa [38]. They reported that NAA10 binds to p21-activated kinase-interacting exchange factor (PIX) to interrupt its downstream Rac1/Cdc42 pathway. Consequently, blockade of the Rac1/Cdc42 pathway results in the inhibition of cancer cell metastasis. This process does not appear to be related to the acetyltransferase activity of NAA10. In BCa, NAA10 inhibits cancer cell migration by blocking the signal transducer and activator of transcription 5α (STAT5α), regardless of acetyltransferase activity [51].

Whether NAA10 retains the opposite functions in different cancer types or under distinct conditions requires further study. Additionally, it is worthwhile to investigate the functions of NAA10, be it as an oncoprotein or a tumor suppressor, that could be rendered by cancer specific microenvironment.

## 7. Other NATs in Cancer

In addition to NAA10, other NATs play roles in cancer progression. NatB, NatC, and NatD have been reported as essential components for cancer cell proliferation and survival. The NatB subunits are overexpressed in HCC, and this upregulation is associated with microscopic vascular invasion [57]. Neri et al. showed that NatB silencing blocks tumor formation and proliferation possibly via NatB-mediated Nt-acetylation of cyclin-dependent kinase 2 and tropomyosin [57]. Overexpression of the NatC catalytic subunit NAA30 increases cancer cell viability [58]. Conversely, knockdown of NAA30 induces p53-dependent apoptosis, disruption of mitochondrial function, and reduction of tumorigenic features in cancer cells [59,60,61]. NatD catalytic subunit NAA40 is required for the survival of human colon cancer cells and its depletion induces apoptotic cell death [62]. Ju et al. reported that NatD is frequently upregulated in primary LCa and its expression level correlates with enhanced invasiveness and poor clinical outcomes [63]. They indicated that NatD is a crucial epigenetic modulator of cell invasion during LCa progression. Recently, NatH (NAA80)-mediated actin Nt-acetylation has been reported to regulate cytoskeleton assembly and cell motility and to act as a potential inhibitor of cancer cell motility [64].

## 8. Conclusions and Perspectives

With NAT and KAT activities, NAA10 is a multifunctional protein involved in various cellular activities required for proliferation, differentiation, autophagy, and apoptosis. NAA10 has been found to play crucial roles in tumorigenesis and its upregulated expression has been observed in several cancer tissues, including BCa, LCa, CRC, HCC, PCa, and osteosarcoma [4,5,6,7,8,9]. NAA10 overexpression in these cancer tissues has been found to be significantly correlated with microvascular invasion, lymph node metastasis, survival rate, and recurrence. These oncogenic properties of NAA10 suggest the potential usage of NAA10 as a biomarker for cancer diagnosis and prognosis. However, contrary to the oncogenic roles of NAA10, some studies have reported that NAA10 acts as a tumor suppressor in LCa, BCa, and OSCC [37,38,46,51]. These reports indicate that NAA10 expression is closely associated with improved clinical outcomes, such as longer survival times, smaller tumor size, and less lymph node metastasis.

These contradictory effects of NAA10 may be due to the different signaling pathways in diverse cancer types or the acetylation of different substrates under the cancer specific microenvironment conditions. NAA10 might appear in various kinds of molecular forms in cancer tissues and its physiological roles should be specified depending on the specific molecular form in certain cell type and condition. Therefore, further studies are necessary to investigate the distinguishable functions of NAA10 depending on the protein interactions and modifications specific for cellular context and metabolic stimulation. For instance, new technology, such as single cell analysis, would be helpful to characterize the different functions of NAA10 over space and time in heterogeneous cancer tissues.

Most of current studies for NAA10 introduced in this review measured the expression level of NAA10 using IHC or mRNA analysis. However, one thing that should not be overlooked is that NAA10 is an enzyme. Analysis for its gene expression or protein amount might not reflect the specific characteristics of NAA10 as an acetyltransferase enzyme. Many previous studies support that NAT and KAT activities of NAA10 are essential for its biological roles and that NAA10 function is stimulated by protein modifications rather than gene expression [65]. Therefore, the development of new methods detecting enzyme activity or different modified forms of NAA10 in cancer tissues could be helpful to provide more accurate information.

Since the NAT and KAT activities of NAA10 stimulate cancer development, identifying the substrates and upstream regulators of NAA10 is also important to elucidate its roles in tumor progression. Several studies have elucidated the molecular mechanisms that regulate the expression or catalytic activity of NAA10. Yang et al. showed that miRNA-342-5p and miR-608 inhibit CRC tumorigenesis by targeting NAA10 mRNA for degradation [42]. In addition, phosphorylation appears to negatively regulate the stability or enzymatic activity of NAA10. Phosphorylation of Ser209 and Ser228 residues decreased the stability and activity of NAA10 [25,32]. In contrast to phosphorylation, acetylation of K136 residue enhance the KAT activity of NAA10, promoting proliferation and survival of cancer cells [24,30]. These findings may provide a clue for discovering inhibitors or activators of NAA10 that can be useful to improve therapeutic treatments for cancer patients.

In this review, we provide evidence that NAA10 can be used as a novel target and a potential biomarker for the development of more advanced cancer therapeutics. If used in combination with other biomarkers, NAA10 could provide more precise and accurate therapeutic modalities for cancer patients. Future studies should focus on clarifying the precise roles of NAA10 in tumorigenesis and identifying a new molecular control system of NAA10. In addition, the role of NAA10 in other cancers not mentioned in this review should be explored.

## Figures and Tables

**Figure 1 ijms-21-08010-f001:**
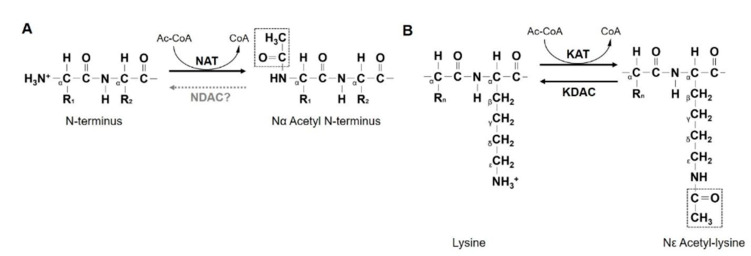
N-terminal and internal lysine acetylation by NAA10. (**A**) NAA10 acetylates the α-amino group of N-terminal residue of new peptides. An acetyl group (dotted rectangle) is transferred from acetyl-CoA to a free α-amino group at the N-terminal. (**B**) NAA10 also acetylates the ε-amino group of an internal lysine residue of the protein. The acetylated lysine can be deacetylated by the lysine deacetylase (KDAC), whereas N-terminal deacetylases (NDACs) have not been reported yet. NAT, N-terminal acetyltransferase; KAT, lysine acetyltransferase.

**Figure 2 ijms-21-08010-f002:**
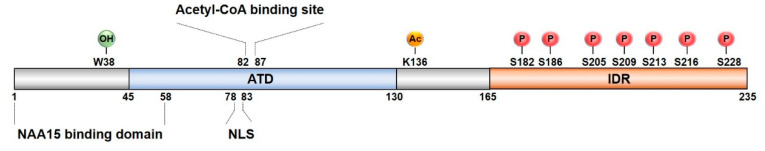
Domain structure of NAA10. The human NAA10 protein consists of 235 amino acids. Acetyltransferase domain (ATD) is located between amino acids 45 and 130, containing acetyl-CoA binding site and nuclear localization signal (NLS). N-terminal region of NAA10 is critical for the binding with NAA15 and C-terminal region is predicted as an intrinsically disordered region (IDR). NAA10 has several post-translational modification sites for phosphorylation, acetylation, and hydroxylation. Ac, acetylation; OH, hydroxylation; P, phosphorylation.

**Table 1 ijms-21-08010-t001:** Clinical outcome in cancer tissues overexpressing NAA10.

Prognosis	Cancer Type	Clinical Outcome	Reference
Overall Survival	Breast cancer	Low	Wang et al., 2011 [4]
		High	Kuo et al., 2010 [37]
		High	Zeng et al., 2014 [51]
	Colon cancer	Low	Jiang et al., 2010 [6]
	HCC	Low	Lee et al., 2018 [19]
	Lung cancer	Low	Lee et al., 2010 [8]
		High	Hua et al., 2011 [38]
	OSCC	High	Zeng et al., 2016 [46]
	Osteosarcoma	Low	Chien et al., 2018 [5]
Invasiveness	HCC	MVI	Shim et al., 2012 [7]
Metastasis	Breast cancer	High	Wang et al., 2011 [4]
		Low	Kuo et al., 2010 [37]
		Low	Zeng et al., 2011 [51]
	Lung cancer	Low	Hua et al., 2011 [38]
	OSCC	Low	Zeng et al., 2016 [46]
	Osteosarcoma	High	Chien et al., 2018 [5]
Recurrence	Breast cancer	High	Wang et al., 2011 [4]
	HCC	High	Lee et al., 2018 [19]
	OSCC	Low	Zeng et al., 2016 [46]

HCC, hepatocellular carcinoma; OSCC, oral squamous carcinoma; MVI, microvascular invasion.

**Table 2 ijms-21-08010-t002:** The role of NAA10 and its regulatory effects on target proteins in cancer progression.

Role	Target Protein	Function	Effect of NAA10	Activity	Reference
Oncoprotein	β-catenin	Proliferation	Acetylation	Activated	Lim et al., 2016 [6]
	Cdc25A	Proliferation	Acetylation	Activated	Lozada et al., 2016 [52]
	AR	Proliferation	Acetylation	Activated	DePaolo et al., 2016 [22]
	PGK1	Autophagy	Acetylation at K388	Activated	Qian et al., 2017 [25]
	Hsp70	Apoptosis	Acetylation at K77	Activated	Park et al., 2017 [53]
	DNMT1	Migration	Interaction	Activated	Lee et al., 2010 [8]
	AuA	Migration	Acetylation at K75/K125	Activated	Vo et al., 2017 [54]
	MMP-2	Migration	Stabilization	Activated	Chien et al., 2018 [5]
Tumor suppressor	TSC2	Autophagy	Acetylation	Activated	Kuo et al., 2010 [37]
	MLCK	Migration	Acetylation at K608	Inactivated	Shin et al., 2009 [55]
	PIX	Migration	Interaction	Inactivated	Hua et al., 2011 [38]
	STAT5α	Migration	Interaction	Inactivated	Zeng et al., 2014 [51]

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
