# Peer review of "NAA10 as a New Prognostic Marker for Cancer Progression"

_ijms, 2020, doi:10.3390/ijms21218010_

Round 1
Reviewer 1 Report
The manuscript by Kim et al. is a comprehensive review on the dual role of NAA10 as oncoprotein or oncosuppressor in tumorigenesis and its potential usage as prognostic marker for different tumor types.
Just a minor point to be clarified:
From par. 2, lines 56-58 I understand that NAT proteins have a prevailing cytosolic localization; however, in par. 3, from line 74, it is referred that NAA10 has a NLS, probably not expected for a subunit of NatA complex. Is it a prerogative of NAA10 or a feature shared by the other NAT isoforms? This could be relevant, given the "epigenetic role" of acetylation.
Reviewer 2 Report
Organisms are complex systems with non-linear functional responses. This means that a linear response is not predictable but can be isolated and highlighted only if we know the space-time location of the metabolic event to which it belongs. The same logic applies to the relationships between the expression of genes and their space-time location. Although the encoding of a gene is the patrimony of the single genomes of all cell types that make up a multicellular organism, their expression is correlated to space-time events generated by the metabolic context of a specific cell that involves functional associations with many genes. Also in this case, to define the correct expression of a gene we must know where and when it is expressed and together with whom. NAA10 is expressed in 238 different cellular types.
A similar logic also applies to the products of genes, to proteins. Fifty years ago, when we showed a protein, we thought of a compact globular organization that expressed a single function, sometimes modulated through allosteric phenomena. So a one-to-one relationship. Today we know that in the protein databases are stored globular proteins (about 40%) but also 10 - 15% of pure IDPs (intrinsically disordered proteins) and 45 - 50% of mixed proteins (disorder + order). All this transforms the one-many relationships expressed by a single IDP, or by a mixed protein, into a complex system. In fact, during or after its expression, this protein undergoes covalent modifications of various kinds (PTM, truncations, splicing) necessary to implement a structure-function relationship of the one-to-one type but specific to the space-time context in which it must operate. The lack of a space-time characterization does not allow us to define the exact physiological role of the protein.
That said, in their review the authors correlate the expression and functional activity of the NAA10 protein to many types of cancer, so much so they can define it as a functional marker for these cancers. In support of their claim, the authors report the data present in the literature.
The review lacks a paragraph that summarizes, but clearly, what is known about the structure and structure-function relationships of this protein. The authors report in detail the functional activity of this protein, which is to induce a post-translational modification (acetylation) in the target proteins, forgetting that NAA10 itself is in turn the subject of numerous port-translational modifications that deeply alter it, through its space-time events.
Protein attributes for NAA10_HUMAN,P41227. Gene expression: The protein is expressed in 238 different tissues/cellular types. Size: 235 amino acids for the coded protein. The gene encodes an N-terminal acetyltransferase that functions as the catalytic subunit of the major amino-terminal acetyltransferase A complex. Alternate splicing results in multiple transcript variants. Quaternary structure: it is a component of the N-terminal acetyltransferase A (NatA) complex composed of NAA10 and NAA15 or NAA16. The complex interacts with HIF1A (via its ODD domain); with NAA50 and with the ribosome. Associates with HYPK when in complex with NAA15; Interacts (via its C-terminal domain) with TSC2; Interacts with IKBKB; Interacts with HSPA1A and HSPA1B.
Amino acid modifications (PTM): The protein shows numerous PTM sites: 2 for Acetylations and 6 for Phospharylations. Post-translational modifications: the protein is cleaved by caspases during apoptosis. Phosphorylation by IKBKB/IKKB at Ser-209 promotes its proteasome-mediated degradation. Autoacetylated at Lys-136 which stimulates its catalytic activity. The protein is also present in different isoforms.
Moreover, the protein shows many binary interactions: 95 physical interactions are reported in BioGrid; 74 interactions in Interact; 21 functional interactions in STRING. This happens because the protein is not globular. Its net charge at pH 7.0 is –9.3 very far from 0±1, characteristic of globular proteins. The protein is an intrinsically disordered protein of mixed type (ordered segments + disordered segments) with a sequence of Janus type 2: collapsed or expanded (because context dependent), and thus in the Structural Phase Plot of proteins, it is located at center of the Boundary Region 2. The presence of PTMs push gradually the protein structure in the region 3 of disordered coils. From this brief presentation we argue that the native polypeptide at neutral pH populates conformational states represented by ensembles of coils and chimeras of globules. When phosphates are gradually added to the phosphorylation sites of the protein, as a PTM consequence, it changes its structural class with a sequential migration to ensembles populated by Coils, Hairpins, or Chimeras of coils. This is a consequence of the covalent modifications that modify the structure of the polypeptide. But all this also means that the protein is present in a myriad of molecular different forms that are all functionally different and specific for different space-time contexts. All molecular forms have their own structural properties and covalent characteristics that are associated with a specific cellular type and metabolic event. I invite the authors to calculate by combinatorics without repetitions how many potential molecular forms a system with 8 sites can produce, two of which have different activities from the remaining six. The number of structures will amaze them.
Thus: What is the molecular form that operates in cancers? In what context does it operate? Are the metabolic context of the different cancers similar? Does the same molecular form always operate in different cancers or are they different molecular forms? Is the function of NAA10 in cancers ascribable to the same molecular form, or are they different molecular forms specific to a certain cancer? The last question is generic, considering that cancerous tissue is heterogeneous from a cellular point of view.
These questions should be answered critically and made available to readers.
But this is not the only issue when dealing with a complex problem in a reductionist way. As an example, this referee has performed a systematic literature review for HCC functional top-genes, selecting 35 papers (2013–2018) from PubMed (excluding the paper cited by the authors) and finding 335 top-genes (often HUB genes) involved into HCC. No paper reported NAA10 associated to HCC top-genes and/or marker genes. This referee found that great part of discovered genes were also redundant (that is, different authors report the same gene many times in different papers, which suggests poor review). I attach the reference list.
The analysis carried out suggests that selecting a gene from the literature to associate it with a cancer is a very random event, which should also be considered based on all the top-genes of that cancer. Discussing a gene alone doesn't make much sense if other authors find many top-genes for that cancer but not yours. This methodological approach is also used to characterize sets of functional top-genes in other types of cancer. There is something wrong with the search for top-genes in cancer. This referee thinks that we have to connect the problem with the use of a reductionist approach to a complex problem. When researchers look for genes/proteins that are functionally involved in cancer, they use bioinformatics analysis platforms that refer to annotated databases. In these systems, curators associate all the novel functions found for a protein/gene with its native form. In this way, we attribute many functions to the native form (which instead are attributes of the covalently modified molecular forms). The molecular forms are structurally and functionally very different from the native molecular form. When it is studied and analyzed, for example through relational systems such as biological networks, the result is that we find the same protein in many functional contexts in which it is not physically present, with wrong attributions. NAA10 can be found in 238 different functional context that experience different space and time events.
Therefore, in this review we have two critical problems:
1) NAA10 has an intrinsically disordered structure of a mixed type and is present in humans in a myriad of molecular forms with different levels of expression and functionality over time and space. Referring to the native protein makes no sense because we attribute functional relations that the protein possesses only as a specific molecular form at certain times and in a specific cellular context. Collapsing all the functional characteristics of the many molecular forms to the native form (which does not actually exist functionally) makes no sense and leads to errors. From these errors we generate flawed lists of genes involved in the various cancers.
2) In the literature, there are an impressive number of top-genes for HCC, but NAA10 seems to have been reported in only one paper out of 35 of the same period.
Thus: What are the real top-genes among the 335? And why NAA10 was not found associated to these top-genes (or HUB genes)?
The review reports a reductionist view of the functional history of NAA10, which instead takes place in a relational context with high levels of functional complexity. We have new technology, such as the single cell genomics that takes into account the space and time in which the functional characterization takes place. Its functional answers are much, much more precise. Bringing back old approaches doesn't make much sense because it pollutes the minds. A non-casual example is HCC, a cancer that after years and years of research with the characterization of a myriad of pseudo top-genes, is still deadly because we cannot cure it by identifying a correct pharmacological target. This happens because the functional data found by system analytics is virtual and should tested in solution with targeted biochemical experiments. Few do it, thus many identified functions are non-existent and misleading.
The review is not publishable under these conditions. It must be rewritten and critically adapted to current knowledge. The task of the writer of a review is not only to report merely facts and events but to place them critically in the current context of scientific knowledge.
Round 2
Reviewer 2 Report
The Authors improved the manuscript. Substantial parts have been revised and are now consistent with the relevant scientific literature. In my opinion the review is publishable.